# Novel Structures of Type 1 Glyceraldehyde-3-phosphate Dehydrogenase from *Escherichia coli* Provide New Insights into the Mechanism of Generation of 1,3-Bisphosphoglyceric Acid

**DOI:** 10.3390/biom11111565

**Published:** 2021-10-22

**Authors:** Li Zhang, Meiruo Liu, Luyao Bao, Kristina I. Boström, Yucheng Yao, Jixi Li, Shaohua Gu, Chaoneng Ji

**Affiliations:** 1State Key Laboratory of Genetic Engineering, School of Life Sciences, Fudan University, Shanghai 200438, China; lizhang1@g.ucla.edu (L.Z.); mandymrliu@gmail.com (M.L.); baoly@fudan.edu.cn (L.B.); lijixi@fudan.edu.cn (J.L.); shaohuagu@fudan.edu.cn (S.G.); 2Division of Cardiology, David Geffen School of Medicine at UCLA, Los Angeles, CA 90095-1679, USA; kbostrom@mednet.ucla.edu (K.I.B.); yyao@mednet.ucla.edu (Y.Y.); 3Shanghai Engineering Research Center of Industrial Microorganisms, Shanghai 200438, China

**Keywords:** *Escherichia coli* type 1 GAPDH, 1,3-diphosphoglycerate, BPG, thioacyl intermediate structure, catalytic process

## Abstract

Glyceraldehyde-3-phosphate dehydrogenase (GAPDH) is a highly conserved enzyme involved in the ubiquitous process of glycolysis and presents a loop (residues 208–215 of *Escherichia coli* GAPDH) in two alternative conformations (I and II). It is uncertain what triggers this loop rearrangement, as well as which is the precise site from which phosphate attacks the thioacyl intermediate precursor of 1,3-bisphosphoglycerate (BPG). To clarify these uncertainties, we determined the crystal structures of complexes of wild-type GAPDH (WT) with NAD and phosphate or G3P, and of essentially inactive GAPDH mutants (C150S, H177A), trapping crystal structures for the thioacyl intermediate or for ternary complexes with NAD and either phosphate, BPG, or G3P. Analysis of these structures reported here lead us to propose that phosphate is located in the “new Pi site” attacks the thioester bond of the thioacyl intermediate to generate 1,3-bisphosphoglyceric acid (BPG). In the structure of the thioacyl intermediate, the mobile loop is in conformation II in subunits O, P, and R, while both conformations coexist in subunit Q. Moreover, only the Q subunit hosts bound NADH. In the R subunit, only the pyrophosphate part of NADH is well defined, and NADH is totally absent from the O and P subunits. Thus, the change in loop conformation appears to occur after NADH is produced, before NADH is released. In addition, two new D-glyceraldehyde-3-phosphate (G3P) binding forms are observed in WT.NAD.G3P and C150A+H177A.NAD.G3P. In summary, this paper improves our understanding of the GAPDH catalytic mechanism, particularly regarding BPG formation.

## 1. Introduction

Glyceraldehyde-3-phosphate dehydrogenase (GAPDH, EC:1.2.1.12) is an important enzyme in the glycolysis process. In the presence of inorganic phosphate (Pi), GAPDH catalyzes the conversion of D-glyceraldehyde-3-phosphate (G3P) to 1,3-diphosphoglycerate (BPG), accompanying the reduction of NAD^+^ to NADH. GAPDH in humans is widely known as a drug target because of its central role in glycolysis and in non-glycolytic processes, such as RNA transport, membrane fusion, and apoptosis [1,2]. In enteropathogenic *Escherichia coli* (*E. coli*), GAPDH as a secreted protein interacts with human plasminogen and fibrinogen [3]. Moreover, the secreted protein may play a role in oxidation and protect bacteria from the host oxidative response [4].

Many structures of GAPDH from eukaryotes [5,6,7], bacteria [8,9], and archaea [10,11] have been reported. Eukaryotic and bacterial structures may have the same catalytic process, which is different from that of archaea. The catalytic process includes oxidation and phosphorylation [5,12,13,14]. In the oxidation step, the thiol of the GAPDH active site (Cys150) attacks the aldehyde carbon of G3P, leading to the formation of a hemithioacetal intermediate. Then, hydride is transferred from the hemithioacetal intermediate to the C4 of NAD^+^ to generate an NADH and thioacyl intermediate [15]. In the phosphorylation step, inorganic phosphate nucleophilically attacks the thioester bond of the thioacyl intermediate to generate BPG.

Two anion recognition sites, “Ps” and “Pi”, bind the C3 phosphate of G3P and inorganic phosphate, respectively [16]. This does not mean that the “Ps” site can only bind the C3 phosphate of G3P. The “Ps” site can also bind inorganic phosphate, and it is well conserved in various GAPDH structures [12,17,18], while the “Pi” site is not. GAPDH contains two “Pi” sites, one being the “classic Pi” originally found in *Bacillus stearothermophilus* (*Bs*GAPDH, 1GD1) [19] and the other being the “new Pi” site, which was first observed in *Thermotoga maritima* (*Tm*GAPDH, 1HDG) [8]. The location of the “Pi” site is related to the loop conformation [15] formed by amino acid residues from 208 to 215, which labels the residents in glyceraldehyde-3-phosphate dehydrogenase type 1 from *Escherichia coli* (*Ec*GAPDH1). Two conformations, named I and II, have been observed for the loop. The most prevalent structural form is conformation I, which is present in all GAPDH structures except the thioacyl intermediate structure, and which is related to the “Ps” site. The alternative loop conformation II, which corresponds to the “new Pi” site, is primarily observed in the thioacyl intermediate structure of *Bacillus stearothermophilus* [20].

The GAPDH “flip-flop” catalytic model is based on the thioacyl intermediate structure proposed [15,20]. At the beginning, the C3 phosphate of G3P binds to the “Ps” site. After hydrogen transfer forms a thioacyl intermediate, C3 phosphate flips to the “new Pi” site. At last, it flops back to the initial site before phosphorylation. Several aspects of the GAPDH “flip-flop” catalytic model are still unclear. The precise site of the phosphate that attacks the thioacyl intermediate to generate BPG is still unknown. Whether the C3 phosphate of G3P will return to the initial site before the phosphorylation step is still unknown. Better knowledge of the structure including BPG and the cofactor NAD^+^ will provide detailed information about the GAPDH catalytic process.

In the thioacyl intermediate structure, the role of the cofactor in the amino acid rearrangement is still unclear. Sébastien Moniot [20] speculated that NADH formation trigged a change in the loop conformation. There is not sufficient structural evidence to prove this surmise. In the thioacyl intermediate reported by Sébastien Moniot, all the subunits contain an altered loop conformation and a cofactor. Cofactors NAD^+^ and NADH cannot be distinguished from the electron density map. It is not clear whether the cofactor is NAD^+^, NADH, or a mixture. If the cofactor is NAD^+^, the new NAD^+^ will re-enter the catalytic domain. This event happens in a phosphorylation step. However, if the cofactor is NADH, this means that hydride transfer has just finished. In the thioacyl intermediate reported by Somnath Mukherjee [21], only the altered loop exists. Therefore, when and how the loop conformation changes are still unclear. The loop conformation change is a very important step in the GAPDH catalytic process. If we know how and when the loop conformation changes, we can precisely stop the catalytic process in pathogenic bacteria. It would be helpful to find a special drug to kill pathogenic bacteria that does not hurt humans. Most pathogenic bacteria obtain energy through glycolysis. Therefore, it is very important to know when amino acid rearrangement happens. Thus, it is necessary to research this “old” but very important enzyme.

In this paper, a series of crystallographic structures of *Ec*GAPDH1 complex are reported. Through analysis two new reported structures of C150S.NAD.BPG, H177A.NAD.BPG thioacyl intermediate, and ternary complexes with phosphate, we propose that inorganic phosphate located in the “new P_i_” site attacks the thioacyl intermediate, leading to BPG generation. Besides that, two new G3P binding forms are reported in this paper. This provides novel insights into substrate binding and product generation and help to elucidate the complete catalytic process of GAPDH.

## 2. Materials and Methods

### 2.1. The Mutagenesis

Mutants C150S, C150A, and H177A were generated by overlap extension polymerase chain reaction (PCR), as described previously [22,23], using the wild-type vector as the template and primers, as shown in Appendix A. The double mutant C150A+H177A was generated using a vector containing C150A as the template and H177A-F and H177A-R as primers. Positive clones were verified by DNA sequencing.

### 2.2. EcGAPDH1 Mutants’ Protein Expression and Purification

Mutants of *Ec*GAPDH1 were purified as previously described [24]. First, we used the Ni-NTA purification method to exclude nonspecific proteins. The protein was further purified by size-exclusion chromatography on a high-resolution HiPrep^TM^ 26/60 sephacryl S-100 column (Amersham Pharmacia Biotech, Uppsala, Sweden) with an elution buffer containing 400 mM NaCl and 40 mM Tris–HCl pH 8.0 as the mobile phase. The purified protein was concentrated and changed to a low salt buffer including 4 mM NaCl and 5 mM Tris-HCl at pH 8.0 using the centrifuge method in 30 KDa Amicon^®^ Ultra-15 Centrifugal Filter Units. The purity of *Ec*GAPDH1 was analyzed by 12% SDS-PAGE. The concentration of the protein was determined by the Bradford assay.

### 2.3. Enzyme Activity

The activity of *Ec*GAPDH1 mutants was measured by monitoring the change in absorbance at 340 nm because of NADH formation [25]. Assays were performed in 96-well plates on a DU800 UV-visible spectrophotometer (Beckman, USA) machine at 50 °C. The standard assay was carried out in a 200 µL solution containing 40 mM triethanolamine, 50 mM K_2_HPO_4_, and 0.2 mM EDTA at pH 10.0 [24]. The reaction was started by adding 0.022 µg protein/100 µL sing the boiled denatured protein as the blank control. Every experiment was repeated at least three times.

### 2.4. Crystallization and Data Collection

All crystals reported in this paper were grown at 20 °C using the hanging-drop vapor diffusion technique by mixing 1.2 µL of protein solution (30 mg/mL) with 1.2 µL of reservoir solution and equilibrating it against 500 mL of reservoir solution. The compositions of the precipitant solutions used to obtain protein crystals are shown in Appendix A. The proteins of *Ec*GAPDH1 and mutants for ternary complexes were pretreated with 50 mM of Na_3_PO_4_, or 2 mM of D-G3P co-crystallization to obtain the structures of ternary complexes with phosphate or G3P, respectively. C150S was reacted with 2 mM of NAD^+^ and 2 mM of D-G3P for 5 min to obtain the thioacyl intermediate. C150S and H177A were pretreated with 2 mM NAD^+^, 2 mM D-G3P, and 50 mM Na_3_PO_4_ for 0–5 min to obtain C150S.NAD.BPG and H177A.NAD.BPG.

These crystals were soaked in a cryo-protectant solution consisting of 70–75% reservoir solution and 25−30% PEG1000 or 30% PEG400. Then, the crystals were flash cooled under a nitrogen stream. Diffraction data were collected at the BL17U/18U/19U beamlines of the Shanghai Synchrotron Radiation Facility (Shanghai, China) [26]. X-ray diffraction data were indexed, integrated, and scaled with HKL-2000 [27] or HKL-3000 [28].

### 2.5. Structure Solution and Refinement

The structures of the mutants of *Ec*GAPDH1 were determined by molecular replacement methods using the crystal structure of wild-type *Ec*GAPDH1 (PDB code: 7C5F) [24] as the search model. Model building was carried out by Coot [29] and refined by Phenix [30,31] and Refmac5 [32]. The result of the refinement was assessed using the values of *R_work_* and *R_free_*. The stereochemical quality of the model was evaluated by ProCheck [33]. The graphics were visualized using PyMOL [34].

### 2.6. Secondary Structure-Based Align

Using the DALI server and PDB bank to search for structures of GAPDH, it was found that it has over 100 structures. Many of these structures were from the same species. In the *Homo sapiens* liver, mutants and mutant binding with different substrates were found. When secondary structures aligned, we considered the wild type, mutants of GAPDH, and GAPDH with different substrates as one structure. We randomly chose two structures from bacteria, archaea, and humans. Then, we used the server Clustal Omega for sequence alignment. After that, we used ESpript 3.0 to perform secondary structure alignment [35].

## 3. Results

### 3.1. Protein Purification, Biochemical Properties, and Overall Structure of EcGAPDH1 Mutants

We surmised the activity of amino acids Cys150 and His177 through secondary structure alignment of multiple referenced organisms [15,16] (Figure 1A). We found that cysteine and serine had similar structures. The only difference was in the side chain, where the serine has hydroxyl while cysteine has sulfhydryl. In order to see whether Cys150 and His177 could affect enzyme activity and structure, we obtained mutants H177A, C150S, C150A, C150A+H177A, and C150S, whose structures were the most similar to that of the wild type.

Mutants of *Ec*GAPDH1 exhibited more than 95% purity after Ni-NTA affinity and gel-filtration chromatography (Appendix A), as estimated by the Quantity One software (Bio-Rad, Irvine, CA, USA).

The activity of *Ec*GAPDH1 mutants was measured in the buffer with 40 mM triethanolamine, 50 mM K_2_HPO_4_ at pH 10.0, and 0.2 mM EDTA at 50 °C. The results showed that the enzymatic activity of C150A and C150A+H177A was abolished. The enzymatic activity of wild-type *Ec*GAPDH1 was 3.84 × 10^4^ times higher than that of C150S and 8.13 × 10^3^ times higher than that of H177A (Figure 1B). These results indicate that C150 and H177 are essential for *Ec*GAPDH1 activity. The apparent K_M_ value of the wild-type *Ec*GAPDH1 enzyme for D-G3P was estimated to be 0.73 mM, the V_max_ was 0.272 mM/min, and the K_cat_ was 224,926 min^−1^. The K_M_ value of *Ec*GAPDH1 was similar to that of *Cryptosporidium parvum* GAPDH (*Cp*GAPDH) [16]. However, the Vmax was 171 times higher than that of *Cp*GAPDH. This means that *Ec*GAPDH1 may have a higher enzymatic activity than *Cp*GAPDH.

The crystallography statistics for data collection and structure refinement are summarized in Table 1 and Appendix A. The *Ec*GAPDH1 crystals belonged to the tetragonal space group P4_1_2_1_2. All the structures reported in this paper have four subunits: O, P, Q, and R. Each subunit contained two domains: the C-terminal catalytic domain and the N-terminal NAD^+^-binding domain. Catalytically active residues C150 and H177 belong to α5 and β9, respectively (Figure 1C). Superimposing the backbone atoms from the structures of C150S.NAD, C150A.NAD, H177A.NAD, and C150AH177A.NAD on WT.NAD, it was found that the RMSDs were 0.134 (1266), 0.182 (1135 atoms), 0.135 (1247 atoms), and 0.122 Å (1219 atoms), respectively (Appendix A). This indicated that mutants have a very similar structure to the wild type. From the B factor distribution of C150S.NAD, it was found that the R subunit had a higher B-factor than the other three subunits. This indicates that the R subunit has a relatively high vibrational motion. The calculated surface electrostatic potential of C150S.NAD was negatively charged (Figure 1E).

Amino acids Arg11, Ile12, Asp34, Arg78, Ser120, and Asn315, which form hydrogen bonds with NAD^+^, are highly conserved in WT and the mutant binary complex structure of *Ec*GAPDH1 (Appendix A).

### 3.2. Inorganic Phosphate Binding: “New Pi” and “Ps” Site

A structural comparison of the ternary complex showed that the “Pi” site in C150S.NAD.PO_4_ (7C5G) and WT.NAD.PO_4_ (7C5H) is 0.1 Å away from the “new Pi” site of *Tm*GAPDH and 3.2 Å away from the classical “Pi” site of *Bs*GAPDH (Figure 2D). These results indicate that the “Pi” site of *Ec*GAPDH1 is the “new Pi” site.

In WT.NAD.PO_4_ (7C5H) and C150S.NAD.PO_4_ (7C5G), each R subunit has two anion recognition sites, “Ps” and “new Pi”, and each of the other subunits (O, P, and Q) harbor only one “Ps” site (Figure 2A). An analysis of the B factor distribution in WT.NAD.PO_4_ found that the R subunit has a higher B factor than the other three subunits (Appendix A). This may be one reason why only the R subunit has two phosphate-binding sites. This result also indicates that the “Ps” site has a higher binding affinity for phosphates than the “new Pi” site. In addition, Chakrabarti [39] and Copley [40] reported that an arginine residue enhances the “Ps” site’s affinity for anions. In light of these reports, the amino acid residues that interact with “Ps” and “new Pi” sites were analyzed. The “Ps” site was found to be stabilized by hydrogen bonds with the side chains of Thr180, Thr182, Arg232, and 2′-OH from the ribose of NAD^+^ (Figure 2C). The “new Pi” site was found to interact with the side chains of Thr209 and Ser149, in addition to the side chain hydroxyl and main chain nitrogen of Thr151 (Figure 2B). The presence of Arg232 in the “Ps” site, not the “new Pi” site, may explain the higher binding affinity of the “Ps” site for phosphate.

In C150A.NAD.PO_4_, all the subunits have two phosphate-binding sites. Phosphate in the “new Pi” site is around the catalytic amino acid Cys150. In C150A.NAD.PO_4_, all the subunits have two phosphate-binding sites. Phosphate in the “new Pi” site is around the active amino acid Cys150. This phosphate will lose the occlusion from the side chain of cystine after mutation. Secondly, the catalytic cavity increased almost 6 Å^3^ after the mutant cysteine to alanine, which allows the “new Pi” site to occupy the phosphate [41].

### 3.3. The Route Followed by G3P to Enter the Catalytic Domain

Although multiple structures of GAPDH binding with G3P have been reported [15,42], how G3P accesses the reactive center remains unclear. Two novel G3P-binding forms were observed in the ternary complexes WT.NAD.G3P and C150A+H177A.NAD.G3P. In these structures, the C3 phosphates of G3P are all located in the “Ps” site, while the “new Pi” site remains unoccupied. G3P binds to the open pocket with a positive charge (Appendix A).

In WT.NAD.G3P (7C5P), G3P binds to all subunits in a completely inverted manner compared to *Tm*GAPDH (C151S.NAD.G3P, 3KV3). In these two structures, the phosphate of G3P is all located in the “Ps” site and the main difference between them is the orientation of O1. O1 faces NAD^+^ in WT.NAD.G3P, whereas it faces the active amino acid cysteine in *Tm*GAPDH. In WT.NAD.G3P, O1 of G3P forms hydrogen bonds with O1N of NAD^+^ and the amide nitrogen of Thr180. O2 is stabilized by hydrogen bonds with O3 and O1A of NAD^+^. Additionally, the phosphate of G3P interacts with the Oγ of Thr182 and Thr180 and the NH1 of Arg232. This binding state is presumably the initial state of G3P after it enters the catalytic center. This is the first presentation of G3P binding in this specific manner (Figure 3A).

In C150A+H177A.NAD.G3P (7C5M), the G3P-binding modes of the O, P, and R subunits are distinct from those of the Q subunit (Figure 3B). There are two main differences between them. First, in the P subunit of C150A+H177A.NAD.G3P, the oxygen atom O1P of the C3 phosphate forms hydrogen bonds with the Oγ atoms of Thr182 and Thr180, as well as with NH1 of Arg232. However, it is the O3P oxygen atom that forms hydrogen bonds with the same amino acid in the Q subunit. The second difference is the orientation of O1 and O2 in G3P. In the P subunit (Figure 3D), O1 of G3P forms hydrogen bonds with Nε of His207, and one water molecule, while O2 of G3P forms hydrogen bonds with water. However, in the Q subunit, the O1 of G3P forms hydrogen bonds with Oγ and the amide nitrogen of Thr151, while the O2 of G3P interacts with the amide nitrogen of Ala150. Oxygen atoms of O4P and O2P form water with the same hydrogen in all subunits, whereas O2P interacts with the O2D of NAD^+^ and water.

G3P mainly appears in hydrate (gem-diol form) and aldehyde form in solution. The unbiased 2mFo-DFc omit maps are contoured at 1.0 σ for the G3P of WT.NAD.G3P (7C5P) and the P, Q, and R subunits of C150A+H177A.NAD.G3P (7C5M), clearly showing that G3P is in an aldehyde form (Appendix A). The G3P of the O subunit of C150A+H177A.NAD.G3P is present in hydrate form (Figure 3C). In C150S.NAD.G3P (7C5K), G3P binds the P subunit in aldehyde form (Appendix A), whereas it is present in the hydrate form in the O, Q, and R subunits (Appendix A). The driving forces that stabilize these two distinct forms are different. In C150S.NAD.G3P, two hydrogen bonds are formed between O1 of G3P and Oγ of Thr209, while the O2 of G3P with the Nε of His177 strengthens the aldehyde form. However, these bonds do not exist in the hydrated form. Upon the superimposition of the above structures of GAPDH on G3P, we can infer that before G3P reacts with the catalytic amino acid Cys150, the C3 phosphate of G3P is in the “Ps” site and binds with the amino acids Thr180, Thr182, and Arg232. Several factors cause the G3P-binding form to be different in the above-mentioned structure. Firstly, in the ternary complexes of *Ec*GAPDH1, the crystal condition includes polyethylene glycol 1000 (PEG1000), which is helpful to stable G3P through noncovalent binding. The acylation rate in the crystal condition is slower than in the aqueous solution [15]. This may be one reason why we can get different kinds of G3P-binding forms. Secondly, G3P will lose the occlusion of the side chain of C150 and H177 after mutation. Then, mutant ternary complexes can move more freely than in the wild-type one. We compared the G3P-binding form in the wild type and mutants, and found that after mutation, the catalytic cavity is increased almost 10 Å^3^ in C150A+H177A [41].This is helpful for G3P to move. Overactive G3P may be one factor that causes the mutants to have a lower enzyme activity than the wild type. We speculated that G3P may be in a state of constant movement and remains fixed at the “Ps” position after it enters the catalytic domain (Figure 3E). The question of how G3P enters the catalytic domain is explored in the discussion.

### 3.4. Structure of the Thioacyl Intermediate

In the wild-type thioacyl intermediate (7C7K), only the Q subunit contains NADH, which has a complete NADH electron density map. G3P binds to the reaction site in all four subunits, which have very similar interactions. An unbiased difference map of the interaction of G3P with Cys150 is shown in Appendix A. The conformation of G3P is stabilized by one thioester bond formed by the C1 of G3P with Sγ of Cys150 and numerous hydrogen bonds. The oxygen atoms of C3-phosphate interact with the side chains of Ser149, Thr151, His177, and Thr209; the main chain nitrogen atoms of Thr151 and Gly210; and three water molecules. In the Q subunit, the O2 of G3P interacts with the Nε of His177 and N7N of NADH. However, G3P fails to interact with N7N of NADH in the other three subunits (Figure 4). These results reveal the fundamental reason why the O and P subunits bear no NADH (Figure 4A) while the R subunit has a partial NADH. The R subunit around the active amino acid has an electron density map that looks like the pyrophosphate part of NADH. When the R subunit was aligned to the Q subunit, it was found that this product can perfectly overlay NADH (Figure 4B,E). In the crystal and purification conditions of *Ec*GAPDH1, we did not add any products related to pyrophosphate (Figure 4D). Therefore, we think that the electron density is one pyrophosphate fragmentation of NADH.

The analysis of the Q subunit of C150S.NAD.G3P and the O subunit of the thioacyl intermediate revealed that there are three differences. First, O1 of G3P is opposite to His177 in terms of its thioacyl intermediate structure, while it is oriented towards His177 in C150S.NAD.G3P. Second, the phosphate of G3P of the thioacyl intermediate is positioned in the “new Pi” site, while it is in the “Ps” site in C150S.NAD.G3P (Figure 4F). Third, the loop that contains amino acid residues 208–215 is in conformation I in C150S.NAD.G3P, while it has two conformations in the thioacyl intermediate.

In the Q subunit of the thioacyl intermediate, the loop has two conformations: conformation I and conformation II (Figure 4D). However, the loop in the other subunits (O, P, R) only exists in conformation II. Moreover, the Q subunit contains a complete coenzyme NADH, whereas the R subunit contains a partial NADH (pyrophosphate), and the O and P subunits contain no NADH at all (Figure 4E). Considering these results, we speculate that it is not NADH release that causes a change in the loop conformation but that the change in the loop induces the release of NADH. Through this structure, we can also see that NADH release is not associated with the flipping of the C3-phosphate group back to the initial site.

### 3.5. BPG Binding and Release in EcGAPDH1

Until now, the structure of BPG bound to GAPDH has not been reported. Researchers have previously only studied the structure of a BPG analog bound to GAPDH from *Trypanosoma cruzi* (*Tc*GAPDH, 1QXS) [17]. The BPG analog used in the previous study contains an extra carbon and is in the S conformation, whereas BPG exists in the R conformation in nature (Figure 5A,B). The structure of ternary complexes H177A.NAD.BPG and C150S.NAD.NPG provides further information on the driving force behind the phosphorylation step and the mechanism of the nucleophilic attack of inorganic phosphate on the carbonyl group of the thioacyl intermediate to generate BPG.

In the structures of H177A.NAD.BPG (7C5Q) and C150S.NAD.BPG (7C5R), the coenzyme NAD^+^ is present in all four subunits. In addition, the O subunit of C150S.NAD.BPG and the P subunit of H177A.NAD.BPG contain an unexpectedly large electron-dense cluster near the active site and NAD^+^ (Appendix A), which was attributed to the product BPG. BPG in C150S.NAD.BPG and H177A.NAD.BPG holds refined occupancy factors of 0.85 and 0.8, respectively. Upon the superimposition of the ternary complexes C150S.NAD.BPG and H177A.NAD.BPG with their corresponding binary complexes, the RMSD values are 0.104 (1197 atoms) and 0.168 Å (1235 atoms), respectively. These results indicate that the overall structures of H177A.NAD.BPG and C150S.NAD.BPG are like those of its binary complex, except for the presence of the product.

The structure of C150S.NAD.BPG likely represents a state in which the enzymatic reaction has attained completion. The oxygen atoms O6P and O7P of the C3 phosphate group form hydrogen bonds with Oγ of Thr182 and Thr180, NH1 of Arg232, and O2D of NAD^+^. Oxygen atoms in the C1 phosphate group are stabilized by numerous hydrogen bonds. O1P forms hydrogen bonds with Oγ of Thr151 and Nε of His177. O2P interacts with the oxygen atom (O7N) of the nicotinamide group of NAD^+^ and Nε of His177. O3P and O4P form hydrogen bonds with the amide nitrogen of Ser150 and with the amide nitrogen of Ser150 and Thr151, respectively. P1 forms a hydrogen bond with Oγ of Ser150. Moreover, O1 forms hydrogen bonds with the side chain of Thr209 (Figure 5B).

BPG is observed in a released state in H177A.NAD.BPG. The oxygen atom of O5P of C3 phosphate forms hydrogen bonds with NH1 of Arg232 and with Oγ of Thr182 and Thr180. O6P and O7P form hydrogens with 2′-OH of NAD^+^ ribose and Oγ of Thr182 and water, respectively. The oxygen atom O3P of the C1 phosphate is stabilized by three hydrogen bonds interacting with the amide nitrogen of Gly210 and with H_2_O 551 and H_2_O 689 (Figure 5C).

Upon the superimposition of BPG and the BPG analog structures, we observed that the C3 phosphate is present in the “Ps” site (Figure 5D). The superimposition of the structures of WT.NAD.PO_4_ with C150S.NAD.BPG revealed that P1 of BPG is 3.0 Å away from the central phosphorus atom of the “new Pi” site but 4.5 Å away from the “Ps” site (Figure 5E). The superimposition of the structures of WT.NAD.PO_4_ and the thioacyl intermediate showed that the carbonyl carbon of the thioacyl intermediate is 4.32 and 6.13 Å away from the central phosphorus atom of phosphate in the “new Pi” site and “Ps” site, respectively (Figure 5F).

## 4. Discussion

### 4.1. Structural Comparison of GAPDH from E. coli with Other Species

The structural similarity analysis was conducted using the DALI server [43] search, which showed that *Ec*GAPDH1 has structural homologies to the previously reported GAPDH, including GAPDH from human placenta (PDB code: 1U8F, *Hu*GAPDH) [37] and testis (PDB code: 5C7L, *Hu*GAPDH) [38], from the archaea *Saccharolobus solfataricus* (PDB code: 1B7G, *Ss*GAPDH) [18], and from the bacteria *Chlamydia trachomatis* (PDB code: 6OK4, *Ct*GAPDH) [36] and *Staphylococcus aureus* MRSA252 (PDB code: 3HQ4, *Sa*GAPDH) [15]. A comparison of the primary sequence and structure between *Ec*GAPDH1 and the other GAPDH from bacteria and humans found that the sequence identity ranged from 44% to 56% and that the RMSD values of the C_α_ atom are 1.3 (for 328 residues in *Hu*GAPDH-5C7L), 1.2 (for 328 residues in *Hu*GAPDH-1U8F), 1.4 (for 331 residues in *C**t*GAPDH), and 1.0 Å (for 331 residues in *Sa*GAPDH), respectively (Figure 1A and Appendix A). However, regardless of its primary sequence or structure, GAPDH from archaea is very different from the GAPDH reported from humans and bacteria. The analysis of the structures of *Ec*GAPDH1 with *Ss*GAPDH found that the RMSD value of C_α_ atom is 3.1 Å (for 276 residues, z score 24.9). The primary identity similarity between them is very low in archaea; the most similar one to our *Ec*GAPDH1 is *Ph*GAPDH from *Pyrococcus horikoshiithe* (PDB code: 2CZC, to be published).Even though, the primary sequence has only a 0.33 identity query that covers only 0.13 of the total sequence. The RMSD value of the C_α_ atom is 2.8 Å (for 274 residues, Z score 25.8) between *Ec*GAPDH1 and *Ph*GAPDH.

Superimposing *Ec*GAPDH1 and the other GAPDH from different species, except from archaea, the result revealed that the overall structure had similar features and that the main differences existed in the loop regions, especially S-loop, which was reported to have a function related to NAD^+^ entering the catalytic domain [44]. In these structures, all the active amino acids are Cys150, and the catalytic-related amino acid is His177 when using the *Ec*GAPDH1 sequence number to label it (Figure 1A).

The analysis of the NAD^+^-binding site in *Ec*GAPDH1 and GAPDH from the other species, especially *Homo sapiens*, found that several amino acids are highly conserved: Arg11, Ile12, Asp34, and Ser120. However, one amino acid was different. The hydrogen bonds between the N atom of the adenine of NAD^+^ and the amino acid in position 78 in *Ec*GAPDH1 is lysine, but it is arginine in humans [24] (Appendix A).

Very few papers have reported on a phosphate-binding site in *Homo sapiens*. We found one structure from human sperm GAPDH (*Hs*GAPDH, PDB code: 3H9E) [5]. The analysis of the primary sequence and structure of *Ec*GAPDH1 and *Hs*GAPDH found that the sequence identity between them is 45% and the RMSD value of the C_α_ atom is 1.3 Å (for 328 residues in *Hs*GAPDH-3H9E). The main difference exists in the loops (Figure 6B). Further analysis of the hydrogen bonds between amino acids and phosphate in “Ps” and “new Pi” found that several amino acids Thr180, Thr182, and Arg232 that stabilize the “Ps” site phosphate, and His177, Ser149, Thr151, and Thr209, which stabilized a “new Pi” site phosphate, are conserved (Figure 6C,D). However, in *Hs*GAPDH, the “new Pi” phosphate has one more hydrogen bond formed between G284 and the phosphate (Figure 6D). This hydrogen bond may be one of the factors that caused the loop, formed by amino acids 208 to 215, to change from conformation I to conformation II (Figure 6A). However, in *Ec*GAPDH1, the loop is still in conformation I. Only in the thioacyl intermediate is the loop in conformation II. This is different from the situation in humans.

### 4.2. GAPDH Catalytic Process

In this study, we provide a series of *Ec*GAPDH1 structures to shed light on the catalytic mechanism of GAPDH. The overall structure analysis showed that the catalytic domain is an open pocket with positive charges.

The catalytic process of GAPDH is as follows: After superimposing our reported different G3P-binding forms in *Ec*GAPDH1 with those reported in other research, we speculated on the process which G3P uses to accesses the catalytic domain. Firstly, G3P enters the open pocket randomly or with the help of a coenzyme. Based on the binding forms of G3P in WT.NAD.G3P, we are more inclined to believe it happens in the latter way. When the C3-phosphate of G3P is in the “Ps” position, G3P will rotate around the “Ps” position until C1 of G3P is close to the catalytic amino acid Cys150 (Figure 7A,B).

Next, the thiol group of Cys150 executes a nucleophilic attack on the carbonyl carbon (C1) of D-G3P to form a hemithioacetal intermediate. This process is rapid, so we could not obtain the structure of this intermediate. Right after that, C1 of the hemithioacetal intermediate transfers a hydride ion to the nicotinamide ring of NAD^+^ to form NADH and a thioacyl intermediate, causing a conformational change: (1) the O1 atom moves away from His177, (2) the C3 phosphate of G3P flips from the “Ps” site toward the “new Pi” site, and (3) the loop formed by amino acid residues 208–215 transforms from conformation I to conformation II (Figure 7D,E and Step IV). In the thioacyl intermediate structure, the simulated annealing (2Fo−DFc) map clearly shows that the amino acid rearrangement precedes NADH release. Conformational changes likely occur after the hydride transfer to the coenzyme NAD^+^. Then, the hydrogen bond between C3 phosphate and the 2’-OH group of the ribose adjacent to the nicotinamide group of NAD^+^ is lost. Meanwhile, the thioester bond between C1 of G3P and Sγ of Cys150 is formed. Under these circumstances, the conformation of G3P changes; C3 phosphate forms a hydrogen bond with the side chain of Thr209 and the main chain nitrogen atom of Gly210. This may be one reason that causes the loop to change from conformation I to conformation II. In conformation II, it is easy for NADH to leave the catalytic domain because the interaction between G3P and NADH is lost. Shortly after NADH leaves the catalytic domain, phosphate will enter the catalytic domain to attack the thioester bond of the thioacyl intermediate, leading to the generation of BPG. In this step, it is advantageous for the phosphorylation reaction to take place when the phosphate of the thioacyl intermediate is in the “Ps” site and the inorganic phosphate is in the “new Pi” site. In this state, the attack path will not be occluded by the C2 hydroxyl moiety. Moreover, the central phosphorus atom of phosphate is 4.32 Å away from the carbonyl carbon of the thioacyl intermediate. When the phosphate positions are reversed, the attack path is occluded and the distance between them changes to 6.13 Å (Figure 5F). Furthermore, the structure of C150S.NAD.BPG shows that P1 of BPG is located 3.0 Å away from the central phosphorus atom of the “new Pi” site but 4.5 Å away from the “Ps” site (Figure 5E). Hence, the inorganic phosphate located in the “new Pi” site is the most favorable conformation for the phosphorylation reaction, and this is where a nucleophilic attack on the thioester bond occurs in the process of BPG formation. This also indicates that C3 phosphate flops back to the initial position before the nucleophilic attack on the thioester bond by the inorganic phosphate (Figure 7 Step V). Considering this, we speculated that new NAD^+^ molecules re-entering the catalytic domain are mainly responsible for the loop conformation flopping back to the initial position. When the new NAD^+^ ion is reloaded into the catalytic domain, the hydrogen bond between G3P and NAD^+^ is reformed, which promotes G3P flopping to the initial position (Figure 7 Step IV). At the last step, BPG leaves the catalytic domain. Upon the superimposition of different structures containing BPG, we suggest that BPG leaves the catalytic domain soon after its formation. BPG rotates in the “Ps” site until the C1 phosphoric acid of BPG interacts with amino acids Gly210, Thr209, or O1N of coenzyme NAD^+^, which releases BPG from the catalytic domain (Figure 7 Step VI). After BPG is released, a new G3P molecule enters the catalytic domain, and a new catalytic cycle occurs.

Overall, we found that bacteria and humans have similar substrate and cofactor-binding forms. However, there are still some differences. The structures of bacteria and archaea are very different. This may indicate that GAPDH from bacteria and humans may have the same original features and that it may have the same catalytic process, which differs from that of archaea. *Ec*GAPDH1 has a “new Pi” phosphate-binding site. The amino acids that stabilize the “Ps” phosphate site are the same in *Ec*GAPDH1 and GAPDH from humans. However, in the “new Pi”-binding site, the structure of GAPDH from humans has one more hydrogen bond formed by phosphate and G284 than *Ec*GAPDH1 does. In these structures, the S-loop, which is related to NAD^+^, enters the catalytic domain, and the other loop formed by amino acids 208–215 in *Ec*GAPDH1, which is related to cofactor exchange, is different. In this paper, we also found two G3P-binding forms in *Ec*GAPDH1. Besides these, we found that the inorganic phosphate located in the “new Pi” site attacks the thioester bond of the thioacyl intermediate, leading to BPG generation, which will complete the GAPDH catalytic process. Further work should be carried out to determine and characterize the structure of the hemithioacetal, which will provide further insight into the GAPDH catalytic process.

## Figures and Tables

**Figure 1 biomolecules-11-01565-f001:**
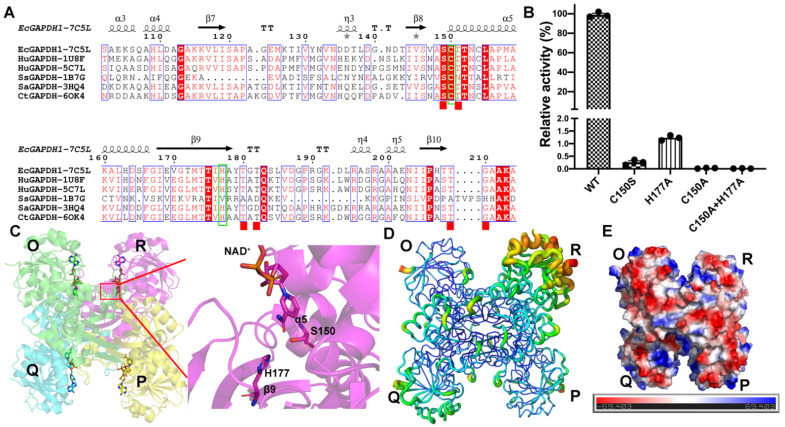
Overall structural and enzyme characteristics of mutants of *Ec*GAPDH1. (**A**) Structural-based alignment of GAPDH from archaea *Saccharolobus solfataricus* (PDB code: 1B7G, *Ss*GAPDH) [18] and bacteria *Chlamydia trachomatis* (PDB code: 6OK4, *Ct*GAPDH) [36], *Staphylococcus aureus* MRSA252 SaGAPDH (PDB code: 3HQ4, *Sa*GAPDH), and GAPDH from human placenta (1U8F) [37] and *testis* (5C7L) [38]. Amino acids that participate in the enzyme catalytic process are presented in the green box. Secondary structure elements included C150S.NAD (7C5L) and wild-type sequences were used for structural alignment. (**B**) Mutant *Ec*GAPDH1 enzyme characteristics. (**C**) Overall structure of C150S.NAD (7C5L). (**D**) The B-factor distribution of C150S.NAD (7C5L). Wider and redder tubing indicates a higher B-factor. (**E**) The structure of the C150S.NAD (7C5L) surface electrostatic potential. Blue represents a positive potential and red depicts a negative potential.

**Figure 2 biomolecules-11-01565-f002:**
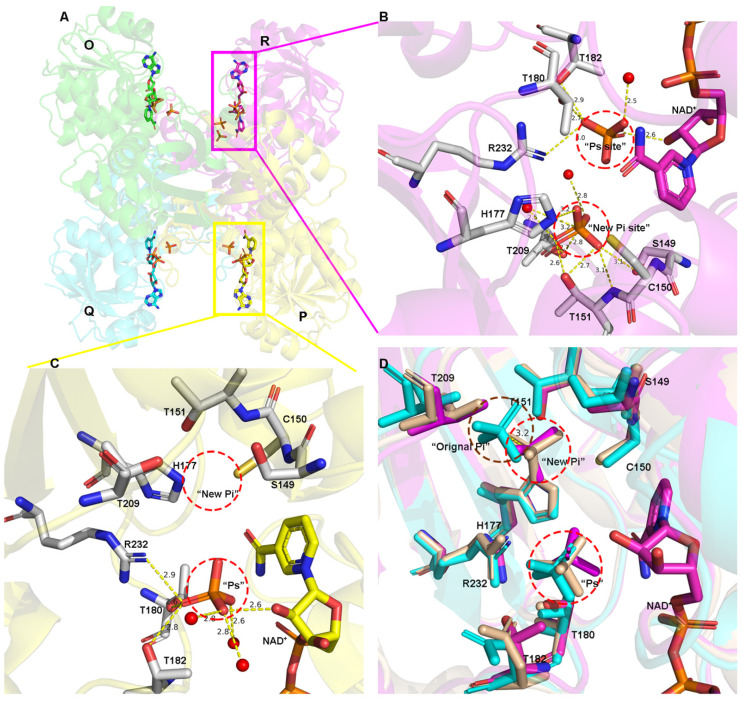
Phosphate-binding sites in *Ec*GAPDH1. (**A**) Phosphate in the overall *Ec*GAPDH1 structure. (**B**,**C**) Phosphate-binding sites labeled as “Ps” and “new Pi” site. B and C represent the R subunit, depicted in magenta, and the P subunit, depicted in yellow, of WT.NAD.PO_4_ (7C5H). The amino acids in contact with phosphate are depicted as sticks and hydrogen bonds are represented by yellow dotted lines. (**D**) Superimposition of the inorganic phosphate and binding amino acid residues of the R subunit of WT.NAD.PO_4_ with the O subunits of *Bs*GAPDH (cyan PDB 1GD1) and *MRSA*GAPDH (wheat PDB 3K73).

**Figure 3 biomolecules-11-01565-f003:**
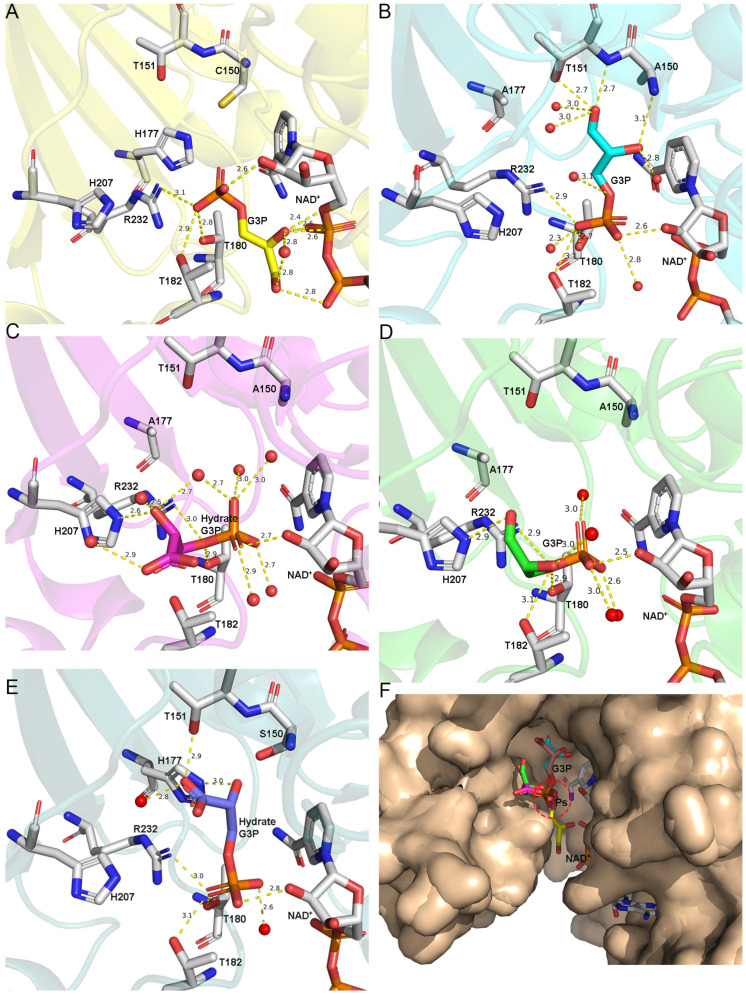
Different binding forms of G3P in *Ec*GAPDH1. The amino acids and water molecules that interact with G3P are shown as sticks with silver carbons and spheres with a red color. Cartoon representation of subunits where hydrogen bonds are depicted by yellow dotted lines. The carbon of G3P is the same as that of the *Ec*GAPDH1 subunit. (**A**) The O subunit of WT.NAD.G3P (7C5P) is shown in yellow. (**B**–**D**) The Q, O, and P subunits of C150A+H177A.NAD.G3P (7C5M), respectively. (**E**) G3P binding in C150S.NAD.G3P (chain Q, PDB 7C5K, tv-blue). G3P in the Q subunit of C150S.NAD.G3P and the O subunit of C150A+H177A.NAD.G3P are in the hydrate form. G3P in the other subunits is in the aldehyde form. (**F**) Superimposition of the O subunit of WT.NAD.G3P (7C5P); the Q, O, and P subunits of C150A+H177A.NAD.G3P (7C5M); and the Q subunit of C150S.NAD.G3P with the P subunit of *Tm*GAPDH (3K73). The subunit is depicted in a wheat-like color, whereas G3P is depicted using sticks.

**Figure 4 biomolecules-11-01565-f004:**
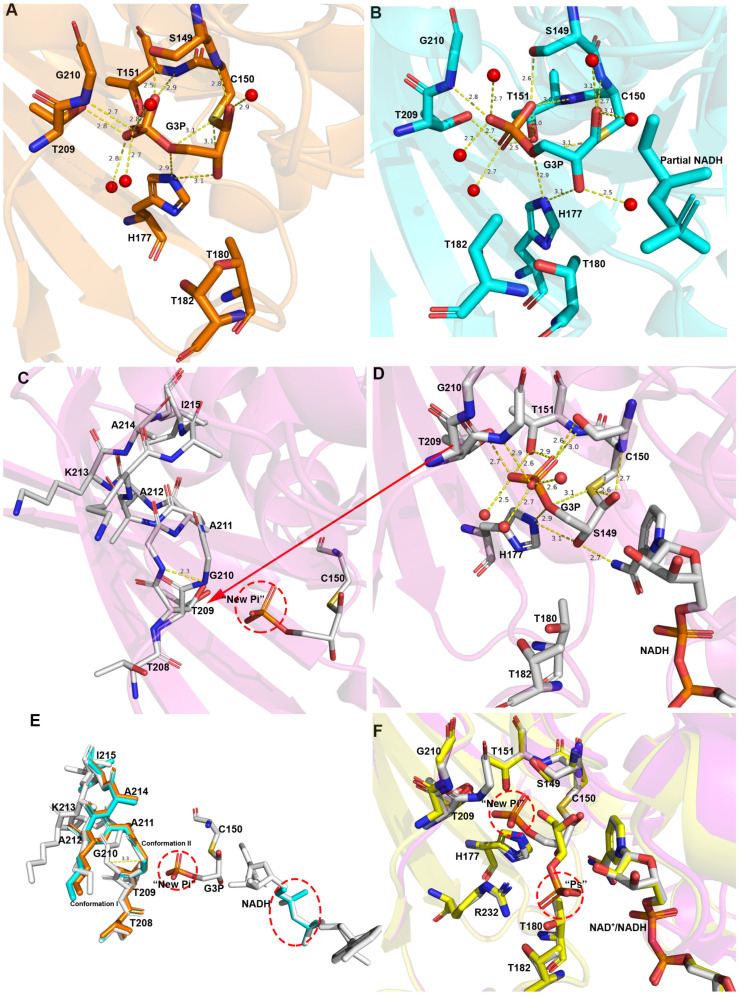
Structure of the thioacyl intermediate of wild-type *Ec*GAPDH1. (**A**,**B**,**D**) NADH and G3P in the P subunit (orange), R subunit (cyan), and Q subunit (magenta) of the thioacyl intermediate of wild-type *Ec*GAPDH1 (7C7K). Amino acids interacting with G3P are shown as sticks and hydrogen bonds are shown as yellow dotted lines. The Q subunit contains complete NADH, the R subunit contains partial NADH, and the P subunit does not contain any NADH. (**C**,**D**) Two different loop conformations composed of amino acid residues 208−215. (**E**) Superimposition of the loop conformation, NADH, and active amino acid interacting with G3P from the above-mentioned structure. The different parts are highlighted in red. (**F**) Superposition of the amino acids interacting with G3P of the Q subunit of the thioacyl complex with the P subunit of C151S.NAD.G3P. The thioacyl complex is shown in magenta, while C151S.NAD.G3P is shown in yellow.

**Figure 5 biomolecules-11-01565-f005:**
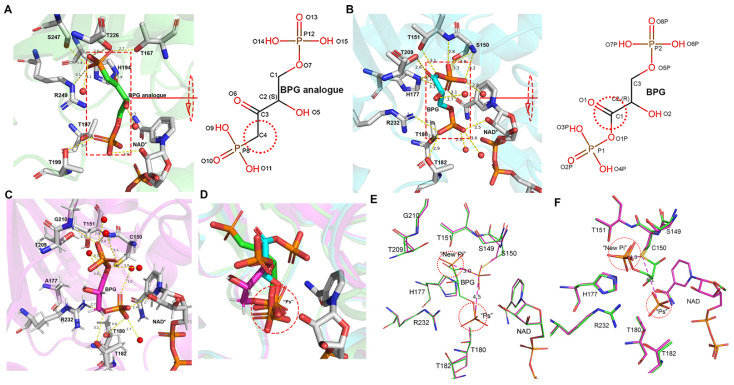
Structure of *Ec*GAPDH1-binding BPG. (**A**–**C**) The network of interactions that stabilize the BPG analog and BPG. (**A**) BPG analog of the A subunit of *Trypanosoma cruzi* GAPDH complexed with an analog of 1,3-bisphospho-d-glyceric acid (*Tc*GAPDH, PDB code 1QXS, green). The left panel of A is a 2-D BPG analog. (**B**) BPG of the O subunit of C150S.NAD.BPG (7C5R, Cyan). The left panel of B is 2-D BPG. (**C**) P subunit of H177A.NAD.BPG (7C5Q, Magenta). (**D**) Superimposition of the above structure. (**E**) Superposition of phosphate, coenzyme, and amino acids interacting with phosphate from the R subunit of WT.NAD.PO_4_ (magenta) with the corresponding amino acids, coenzymes, and BPG of the O subunit of C150S.NAD.BPG (green). (**F**) Superposition coenzyme and amino acids interacting with the phosphate from the R subunit of WT.NAD.PO_4_ with the corresponding amino acids, coenzymes, and G3P of the R subunit of the thioacyl intermediate.

**Figure 6 biomolecules-11-01565-f006:**
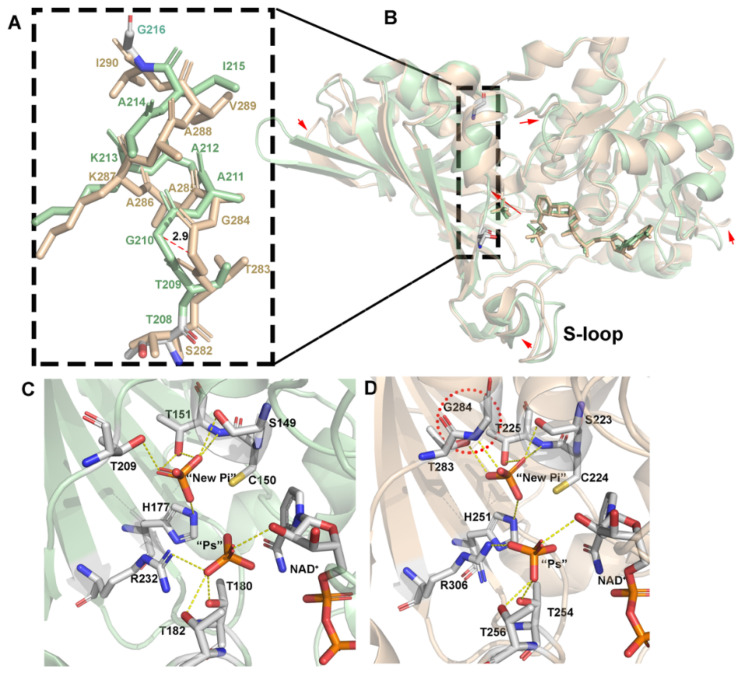
Phosphate binding of GAPDH in humans and bacteria. (**A**,**B**) Loop structure of (**C**,**D**) Phosphate-binding site from the R subunit of WT.NAD.PO4 (7C5H) and O subunit of *Hs*GAPDH (3H9E).

**Figure 7 biomolecules-11-01565-f007:**
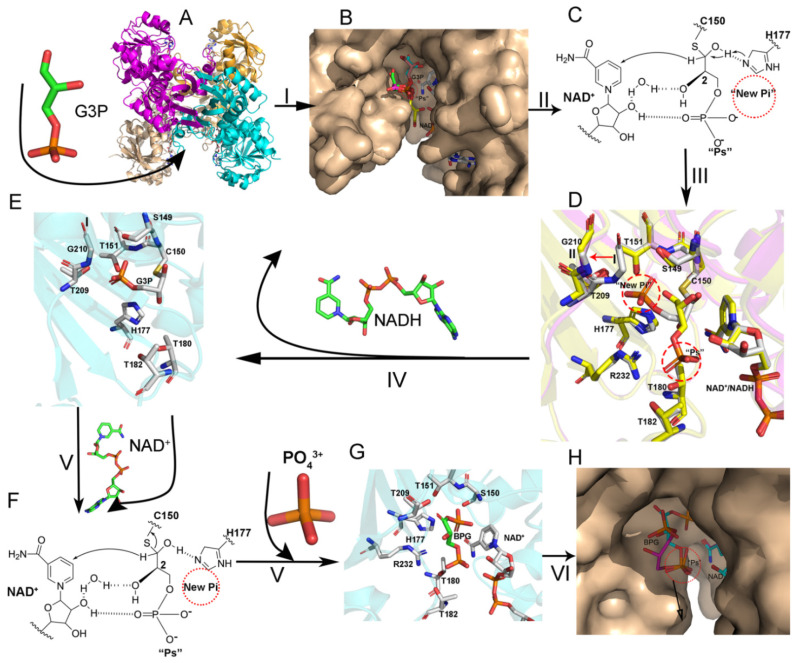
The catalytic process. Step I: G3P enters the catalytic domain. Step II: The thiol group of Cys150 executes a nucleophilic attack on the carbonyl carbon (C1) of D-G3P to form a hemithioacetal intermediate. Step III: Hydrate transfer. Step IV: Coenzyme exchange. Step V: Phosphorylation. Step VI: BPG release. (**A**) The overall structure of *Ec*GAPDH1. (**B**) Superimposition of the O subunit of WT.NAD.G3P (7C5P); the Q, O, and P subunits of C150A+H177A.NAD.G3P (7C5M); and the Q subunit of C150S.NAD.G3P on the P subunit of TmGAPDH (3K73). The subunit is depicted in a wheat-like color, whereas G3P is depicted as sticks. (**C**) Hemithioacetal intermediate. (**D**) Superposition of amino acids interacting with G3P of the Q subunit of the thioacyl complex with the P subunit of C151S.NAD.G3P. The thioacyl complex is shown in magenta, while C151S.NAD.G3P is shown in yellow. (**E**) The R subunit is depicted in cyan of the thioacyl intermediate (7C7K). (**F**) New NAD+ enters the catalytic domain. (**G**) BPG interacts with amino acids and NAD^+^ in the O subunit of C150S.NAD.BPG. (**H**) Superimposition of BPG (cyan) from the O subunit of C150S.NAD.BPG and BPG (magenta) from the P subunit of H177A.NAD.BPG. The BPG is shown as a stick and the subunit is the surface. C and F are the predicted structure, which was drawn by chemdraw.

**Table 1 biomolecules-11-01565-t001:** X-ray data collection and refinement statistics.

	Ternary Complex				Thioacyl Intermediate	Ternary Complex	
	WT.NAD.PO4	WT.NAD.G3P	C150S.NAD.G3P	C150AH177A.NAD.G3P		C150S.NAD.BPG	H177A.NAD.BPG
Data collection							
PDB code	7C5H	7C5P	7C5K	7C5M	7C7K	7C5R	7C5Q
Wavelength (Å)	0.97776	0.97776	0.97776	0.97853	0.97776	0.97776	0.97776
Resolution range (Å) ^a^	50–2.09 (2.13–2.09)	50–2.35 (2.39–2.35)	50–2.68 (2.73–2.68)	50–1.8 (1.83–1.8)	50–1.77 (1.8–1.77)	50–2.31 (2.35–2.31)	50–2.13 (2.17–2.13)
Space group	P4_1_2_1_2						
Unit cell parameters (Å°)	a = b = 89.778, c = 340.953, α = β = γ = 90	a = b = 90.33, c = 341.547, α = β = γ = 90	a = b = 90.378, c = 342.107, α = β = γ = 90	a = b = 89.925, c = 342.379, α = β=γ = 90	a = b = 90.16, c = 345.059, α = β = γ = 90	a = b = 89.398, c = 340.821, α = β = γ = 90	a = b = 89.674, c = 341.795, α = β = γ = 90
Completeness (%)	99.7 (99.3)	100 (100)	100 (100)	100 (100)	99.8 (98.9)	100 (100)	100 (100)
Rmerge (%) ^b^	15.4 (60.4)	19 (62.4)	13.5 (48.7)	11.8 (75.6)	10.9 (60.3)	16.6 (53.5)	15.3 (45.9)
Mean I/δ	13.3 (3.14)	14 (3.71)	15.17 (3.6)	22.5 (3.0)	35.87 (4.75)	12.5 (3.33)	17.75 (5)
No. unique reflections	83,301	60,156	41,105	132,236	139,316	61,720	79,577
Redundancy	13	11.3	12.8	12.6	26.6	16.6	19.4
Refinement							
Resolution range (Å)	39.87–2.09 (2.17–2.09)	48.16–2.35 (2.43–2.35)	48.22–2.68 (2.78–2.68)	44.96–1.8 (1.86–1.8)	48.49–1.77 (1.83–1.77)	47.94–2.31 (2.39–2.31)	48.08–2.13 (2.21–2.13)
Rwork/Rfree (%) ^c^	15.81/20.23	16.31/20.83	16.8/24.1	15.34/19.04	15.37/18.42	15.56/20.70	15.35/19.71
Wilson B-factor (Å^2^)	19.7	30.3	36	22.66	21.38	26.92	21.84
Average B factor(Å^2^)	21.97	32.35	36.51	26.26	24.24	28.94	23.89
B factor (Å^2^)							
Protein	21.59	32.48	36.89	25	23.17	28.72	23.3
Water	25.62	30.73	27.51	34.48	31.99	30.61	28.4
NAD+, PO_4_, G3P, BPG	24.21	31.13	38.8	28.59	32.69	31.52	29.63
Number of NAD/PO_4_/G3P/BPG	4/5/0/0	4/0/4/0	4/0/4/0	4/0/4/0	1/0/4/0	4/2/2/1	4/0/3/1
Number of water molecules	796	575	497	1477	1158	905	950
RMSD bond lengths (Å)	0.007	0.014	0.0114	0.019	0.02	0.015	0.018
RMSD bond angles (deg)	0.984	1.68	1.608	1.89	2.03	1.7	1.88
Ramachandran plot (%)							
Favored	97	96	95	96	97	96	96
Outliers	0.15	0.075	0	0	0	0.075	0
Rotamer outliers (%)	1.1	0.9	1.1	0.55	0.5	0.54	0.82
Clash core	2.64	3.07	1.64	2.26	4.38	2.39	2.06

^a^ Values in parentheses are suitable for the high-resolution shell. ^b^ Rmerge = ΣhΣi|I (h, i) − <I(h)> |/ΣhΣi| I (h, i), where I (h, i) is the average intensity of an individual reflection. ^c^ Rwork is crystallographic for Rfactor and Rfree was calculated as Rwork using 5% of the total reflections excluded from the refinement.

## Data Availability

The coordinates and structural factors were deposited in the Protein Data Bank with the accession codes 7C5J (C150A.NAD), 7C5L (C150S.NAD), 7C5O (H177A.NAD), 7C5N (C150A+H177A.NAD), 7C5H (WT.NAD.PO_4_), 7C5G (C150S.NAD.PO_4_), 7C5I (C150A.NAD.PO_4_), 7C5P (WT.NAD.G3P), 7C5K (C150S.NAD.G3P), 7C5M (C150A+H177A.NAD.G3P), 7C7K (thioacyl intermediate of wild type), 7C5Q (H177A.NAD.BPG), and 7C5R (C150S.NAD.BPG).

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
