# Peer review of "Novel Structures of Type 1 Glyceraldehyde-3-phosphate Dehydrogenase from Escherichia coli Provide New Insights into the Mechanism of Generation of 1,3-Bisphosphoglyceric Acid"

_biomolecules, 2021, doi:10.3390/biom11111565_

Round 1
Reviewer 1 Report
Manuscript summary
This manuscript aims to further elucidate the mechanistic action of GAPDH and ultimately suggests new details on the steps for BPG formation. Specifically, it proposes the location of inorganic phosphate to be at the “new Pi” site during nucleophilic attack of the thioacyl intermediate. Additionally, the study reports that a change of amino acid loop conformation occurs between NADH formation and release. These findings are supported by studying two new crystal structures of Type I GAPDH in E. coli.
General evaluation
The manuscript has clear conclusions that are based on the experimental results and the thought process from data-to-result is clearly described. The study reports the structures of WT and mutant GAPDH, and the conclusions are mainly grounded in preceding literature. By proposing new details on catalytic mechanisms, the manuscript matches the scope of the journal; however, the broader implications EcGAPDH mechanism and activity are not discussed and thus may attract a narrow readership in its current state. Grammar and overall writing should be improved to make the manuscript more understandable. Overall, the manuscript may present a meaningful contribution to the field, but it requires major revisions as detailed below before being reconsidered for publication to the journal.
Major comments:
- The manuscript needs a lot of work to improve its organization, especially in the abstract and introduction. In the present version, the authors go into the details such as the subunits and point mutations of GAPDH enzyme without any introduction, making this manuscript very difficult to read and understand. These sections also seem to list the findings in a rather random order. As there has been a lot of work on GAPDH, the authors need to clarify which parts had been known and which parts are their own findings throughout the whole manuscript.
- Line 162-162: Please provide details for claiming the “high enzymatic activity” of Ec Is CpGAPDH a common enzyme of reference? The introduction mentions identifying differences between GAPDH in pathogenic bacteria versus humans as a motivation for the study. Is there preceding data on human GAPDH activity that you can compare to?
- Lines 88-90: The manuscript should briefly explain the thought process behind choosing point mutations specifically at site 150 and 177. Explain why there was interest in those sites specifically.
- Lines 183-184: The presence of multiple “Ps” sites doesn’t necessarily suggest it has a higher binding affinity than the “new Pi” site. Is there another possible interpretation of why the R subunit is the only one with a “new Pi” site?
- Lines 309-324: Please relate C150S.NAD.BPG and H177A.NAD.BPG by discussing how their different hydrogen bonding may affect why one presents a state of enzymatic completion and the other presents a released state.
- Figure 5: This an important figure and has a potential to become much clearer and better. Explanation of catalytic mechanism is not easy to follow. Some suggestions are as follows. Make it clear in 5A that NAD+ cofactor is already present. Mechanistic diagrams in 5C and 5D should be larger and drawn more clearly. Curved arrows at steps I, IV, and V can have a more uniform curve. Make steps I and VI showing G3P entering in 5A and BPG leaving in 5H visually the same. In the legends, add references to the steps that had been found or described by others, and mark the contributions of the authors by referring back to the structures/figures that the authors obtained/created.
- Discussion: The manuscript would benefit from an additional paragraph or two. Specifically, explain how the claimed high catalytic activity or mechanism of EcGAPDH could influence metabolic driving forces in glycolysis compared to other GAPDH from other organisms that may be less glycolytically active. Second, discuss further mechanistic details that this study’s experiments could not clarify or other studies that could strengthen the proposed mechanistic steps in this manuscript. Lastly, any (new) insights into the regulation of GAPDH arising from the structural information and more detailed mechanisms would be helpful.
- All of these findings are for GAPDH in coli. Please provide some speculation on how conserved you believe this study’s findings might hold across other organisms, specifically on the steps of loop conformation change.
- Additionally, the manuscript requires editing for English grammar and writing. A few grammatical suggestions have been made as an example (see ‘Minor comments’), but thorough revisions by someone with proficiency in scientific communication in English are needed to improve overall understandability.
Minor comments:
- Line 18: “The role of the cofactor in these amino acids rearrange…” grammar is incorrect.
- Line 44: Justify formatting such that there are not so many spaces between text.
- Line 54: Replate “A lot” with “Many”.
- Line 55: Replace “Analyze these structure find” with “Analyses of these structures find”.
- Lines 57-58: Description of the “flip-flop” model is unclear. This sentence either needs a reference to other works describing said model, or it should be omitted.
- Line 68: Replace “About this model, there are still exist some part not very clear.” With “Several aspects of this model are still unclear.”
- Line 73: Remove “have”
- Line 155-156: The “standard reaction” conditions should be briefly described either here or in the methods.
- Figure S1C: Consider using a log-scale graph to show relative activity.
- Table 1: Adjust formatting so that first column categories do not have random large spacing
- Lines 381-382: What is the significance this finding?
Author Response
Hello reviewer,
Thanks for reviewing our paper. You give us so many good suggestions. I am very appreciating your work. This suggestion will make our article better than before. The main change between this revised one and the before one are as following: First I mainly changed the introduction. Now the introduction is following this logic. At the beginning say some general information about GAPDH function and catalytic process. Then compare the GAPDH structure in different species. After that provide why we need to do this research and provide what is new in our research.
Secondly, I add the reason why I mutant amino acids Cys150 and His177. I also increased the description of the overall structure.
Thirdly, in the discussion, I added a structural comparison of GAPDH from E. coli with other species.
It still has a lot of small changes that I will not list here.
The attachment including revising one paper using tracking change, response to reviewer one by one, supplement information, and English edit certificate.
Thanks again for reviewing our paper.
Best wishes,
Chaoneng JI, Li Zhang, and all the other authors

Reviewer 2 Report
The paper aims to fill a knowledge gap in the literature regarding the detailed catalytic
mechanism of Glyceraldehyde-3-Phosphate Dehydrogenase (GAPDH), including the access and
the binding mode of the substrate, glyceraldehyde-3-phosphate (G3P), to the catalytic domain, and
the precise binding site of the inorganic Pi responsible for attacking the thioester bond to generate
the product, 1,3-BPG. In addition, the paper delineates the main factor responsible for the loop
conformational changes within the residues 208-215 in GAPDH during the catalytic reaction. The
premise of the paper is that it provides a full insight of the catalytic mechanism of GAPDH that
can help to distinguish between the bacterial versus human GAPDH in order to design selective
drug against the bacterial GAPDH. Revision is required to improve the clarity and readability of
the paper.
• The abstract needs to be cohesive.
• In general, the introduction lacks sufficient background information about the GAPDH
enzyme. It could be informative to give a more thorough details about GAPDH and explain
the similarity and difference of the enzyme in different strains. Then, focus on the
knowledge gap and how this research will help answer the missing information of GAPDH
catalytic mechanism.
• In the introduction, it would be better to describe the overall ternary structure of GAPDH
before going into details on the recognition sites. Also, line 41 to 52 could be
communicated better.
• Lines from 54 to 66 should be moved early in the introduction.
• The structure of the argument from line 72 needs improvement.
• Emphasize on the novelty of the research and why it was carried out.
• I recommend moving the paragraph from line 90 to 107 to the results section.
• Mention the reason behind mutating the mutated residues C150 and His177.
• In the result section, it is better to explain the overall structure of GAPDH as a
homotetrameric, name each subunit and refer each structure to a figure. Also, label each
subunit in figure1A.
• In general, the data presentation needs improvement. The legends of the figures should be
clear and specific to avoid confusion.
• The discussion needs more elaboration. The author should explain the significance of the
results to by citing previous literature.
• While the study appears to be sound, the language of the manuscript needs improvement
to make it easy to follow.
• Two of references have been duplicated. Reference 16 and 33 and Reference 14 and 15.
Author Response

(The authors gave the same response as above.)

Reviewer 3 Report
The manuscript entitled “Crystal Structure of Type1 Glyceraldehyde-3-Phosphate Dehydrogenase from Escherichia coli Provides New Insight into BPG Generation and Catalytic Mechanism” is a focused on providing novel insight into the catalytic mechanism of Escherichia coli glyceraldehyde-3-phosphate dehydrogenase (EcGAPDH), providing also relevant information regarding 1,3-bisphosphoglyceric acid (BPG) formation. To achieve these mechanistic insights, the authors reported the generation of various EcGAPDH variants, their kinetic analysis and structural characterization in complex with phosphate or substrate, D-glyceraldehyde-3-phsphate (G3P) and the cofactor, NAD+. The information achieved through these investigations allowed the authors to provide meaningful mechanistic insights on this enzyme. I would recommend manuscript publication after addressing the following issues:
- The manuscript is poorly written and deserves substantial English editing; please revise it carefully.
- The target enzyme is named GAPDH, EcGAPDH, EcGAPDH1 throughout the manuscript, please use a unique name to avoid confusion for the reader.
- Introduction, line 40. “Reduction” is more appropriate than “changes”, please modify it.
- The structural and mechanistic background is hard to follow without a supporting figure showing the main structural features reported in this section (e.g. the “Ps” and “Pi” sites, conformations I and II of residues 208-215, etc.) and the scheme of the reaction catalyzed by the enzyme (at least the mechanism known so far).
- Introduction, line 79. “hydrate” should be “hydride”.
- Introduction, lines 87-107. One or two sentences should be added to this part of the introduction explaining the meaning of the mutations performed on the enzyme. The role of the amino acids selected for point mutations should be explained to help the reader in understanding the meaning of the complexes studied by the authors and the results obtained by them.
- Materials and methods, section 2.1. Please add a table (either in the main text or in the Supplementary) showing the sequences of the primers used to generate the EcGAPDH variants.
- Materials and methods, line 130 and Supplementary Table S1. The Supplementary Table S1 shows the composition of the precipitant solutions used for crystallization experiments, thus “The composition of the precipitant solutions used to obtain protein crystals” seems more appropriate than “The crystallization condition” (line 130 of the main text and Table S1).
- Results, lines 155-163. Kinetic data of EcGAPDH and its variants should be displayed in a table in the main text to allow their comparison. The kinetic effects of the mutations can then be described in this section. The role of the mutated amino acids in EcGAPDH and their location in the enzyme structure (e.g. in the active site) should also be reported to allow a better understating of their kinetic effects (see also point 4).
- Tables 1 and S2. X-ray data collection and refinement statistics are reported partially in Table 1 and the remaining part in Table S2, please report all of them in a unique table either in the main text or in the Supplementary. The fifth column of Table 1 only reports “Thioacyl intermediate”, please clarify the complex as for the other columns.
Furthermore, few structures, i.e. NAD, C150A.NAD, H177A.NAD, C150A+H177A.NAD, and C150A.NAD.PO4, reported in the tables are not described in the manuscript, these structures could be either described (in the main text or in the Supplementary) or removed from the work. - Results, section 3.2. The overall structures of the EcGAPDH mutants are not actually described. The authors only asses that there are four subunits, each including the C- and the N-terminal domains. This section should be expanded to provide a general description of the enzyme structure and the relevant information on the overall structures of the mutants, the structural effects of the mutations and the comparison with the native enzyme. Providing this information would allow a better understating of the data reported in the next sections.
- Figure 1. Subunit names should be added to the A panel. In the D panel only the color of the carbon atoms should be changed in the overlapped structures, please modify the figure displayed in this panel.
- Results, section 3.3. The authors report that R subunits have both “Ps” and “new Pi” sites whereas the other subunits has only the “Ps” sites and they deduced a higher binding affinity of phosphate anions for the “Ps” site. Are these sites different among subunits? Is this higher binding affinity supported by other experimental evidence (e.g. Kd or other affinity data)?
Furthermore, the structure has been reported for both wt and C150S mutant, is the mutation affecting the “new Ps” site? According to Figure 1B-C residue 150 is exposed in this site and, at variance with cysteine, serine could contribute to phosphate binding. A further structure determined in complex with phosphate anions is reported in Table S1, C150A.NAD.PO4, does it provide additional data? The description/discussion of this structure should be added to this section.
- Results, line 210. Thr181 should be Thr180 (also in Figure S2B).
- Results, section 3.4. At lines 205-213, the authors observed that the binding of G3P to wt EcGAPDH is inverted with respect to Tm They should try to better justify/explain this difference, is it due to different conditions in which the crystals of the complexes have been obtained? At lines 214-225, the binding of the substrate to the C150A-H177A mutant is described. The authors should discuss the contribution of the mutations to the substrate binding and the comparison with the binding mode observed in the wt. Furthermore, at lines 230-235 the structure of the C150S variant in complex with substrate is introduced but the binding mode is not discussed in details. The authors should describe the main bonds formed within the cavity and the comparison with the other complexes. As reported at lines 235-238 mechanistic insights are deduced by comparing these structures, but the comparison is not actually discussed in the section.
- Results, section 3.5. The presentation of the results is confusing. Please describe first the structure and the differences among subunits in terms of both binding modes and loop conformations, then their comparison with other complexes and the mechanistic insights deduced by the analysis. Furthermore, at lines 276-277 the authors report that the Q subunit contains a complete coenzyme while the P subunit contains a partial NADH, what does it mean? Are the authors suggesting that they are observing NADH fragmentation/degradation in the latter subunit? Please clarify.
- Results, line 291. What does it mean “real structure”? Please clarify.
- Results, section 3.6. The structural comparison should be also extended to the complexes obtained with the substrate and the thioacyl intermediate. This could provide a more detailed view of the changes occurring in the active site during catalysis.
- Atom names/numbering are extensively used for the description of the binding modes throughout the manuscript and they should be also displayed in the figures to allow a better understanding of the reported data.
- The authors should consider supporting the proposed mechanism by molecular dynamic simulations. Furthermore, at lines 362-363 they report that NADH leave the catalytic domain because its interaction with G3P is lost, then, at lines 387-388 they propose that alterations in loop conformation induces the release of NADH. These two sentences seem to suggest different mechanisms, the authors should try to better explain the proposed mechanism and to clarify these concepts.
Author Response

(The authors gave the same response as above.)

Round 2
Reviewer 1 Report
This manuscript is overall improved from the first submission, with the reviewers’ points aptly addressed. With the following additional revisions, this manuscript may be accepted.
- Abstract, line 29. Clearly define the acronym “G3P” for its first use.
- Introduction, lines 103-11. Please restructure paragraph to first briefly introduce the “flip-flop” model, followed by “Several aspects…are still unclear” and details of what precisely is unknown.
- Introduction, lines 82-83. Delete “Detailed information on this can be seen in the discussion.”.
- Introduction, lines 129-132. Briefly include a short (1-2 sentence) summary of what step(s) your reported structures elucidates, to more clearly define the scientific contribution of these structures.
- Results 3.1, lines 818-819. Change to “secondary structure alignment of multiple referenced organisms”.
- Results 3.3, lines 1100-1101. Please clarify what this sentence means/revise.
- Results 3.4, lines 1194-1195. Change “the crystal condition is different” to the exact difference in question.
- Results 3.4, lines 1198. Change “a little bigger” to a more descriptive or quantitative comparison.
- Results 3.6, lines 1545-1547. Change “We only know the structure of the BPG analog bound to…” to “Researchers have previously only studied the structure of a BPG analog bound to…”
- Discussion, line 1763. Further specify “found to be different”. Delete “The detailed information can be seen in [24].” and instead add the [24] citation to the end of the previous sentence.
- Discussion, line 1950. Change “are the main thing causing the loop” to “are what cause the loop”.
- Discussion, lines 2131-2132. Make into same paragraph as the one above it.
- Figure 7 (previously 5). Further improvements could be made to this figure to clearly describe the proposed mechanism. Improve the photo resolution and size of 5C and 5F. Add arrow in 5H showing BPG release similar to 5A. Decrease the size of the molecules NADH, NAD+, and PO4 in steps IV and V. Add appropriate charge to PO4.
Author Response
Dear reviewers,
Thanks for giving me so many good suggestions. I am appreciating for your help. This version does not have a big change compare with before version.
After I accept all the changes from the last version. The new version including some change suggestions by reviewers, and some grammar. The attachment is the “respond to reviewers” one by one.
Thanks again for reviewing and editing our paper.
Best wishes,
Chaoneng Ji, Li Zhang, and all the other authors

Reviewer 3 Report
The authors made substantial efforts in addressing the reviewer’s points and the manuscript is now significantly improved. I would recommend its acceptance after addressing few minor issues, listed below.
- Avoid not explained acronyms, as BPG, in the title.
- Abstract, line 29. Change “G3P” in “D-glyceraldehyde-3-phosphate (G3P)”.
- Introduction, line 41. Correct “embrane” in “membrane”.
- Introduction, line 97. Change “EcGADPH1” in “ coli GADPH1 (EcGADPH1)”
- Introduction, line 97-98. Change the sentence as “Two conformations, named I and II, have been observed for the loop.”
- Introduction, line 97. Change “Escherichia coli glyceraldehyde-3-phosphate dehydrogenase type 1 (EcGAPDH1)” in “EcGADPH1” (see also point 4).
- Materials and methods, section 2.3, lines 623-624. Report the final amount of enzyme in the assay for clarity.
- Materials and methods, section 2.4, line 627. Remove “by co-crystallization” since not all structures are complexes.
- Results, section 3.1, line 820. Change “N-terminal” in “side chain”.
- Results, section 3.1, lines 834 and 834. “-1” should be apex and “max” should be subscript.
- Results, section 3.1, line 820. Change “exist in” in “belong to”.
- Results, section 3.3, lines 1099-1100. Change “active amino acid” in “catalytic”.
- Results, section 3.3, lines 1100-1102. The sentences are not clear, please revise.
- Results, section 3.4, line 1198. The concept “a little bigger” is not specific, please add measurements to support and explain it.
- Discussion, section 4.2, line 1910. “First, G3P enters the catalytic domain.” could be removed since this concept is specified below where it is more appropriate.
- Discussion, section 4.2, lines 1950-1951. Revise “are the main thing causing the loop con-1950 formation to flop back to the initial position.” As “are mainly responsible for the flop-back of the loop to the initial position.”.
Author Response

(The authors gave the same response as above.)
